# Analysis of the Concordance Between the Use of Phenotypic Screening Tests with the β-Lactamase Gene Profile in Selected Gram-Negative Bacteria

**DOI:** 10.3390/antibiotics14121275

**Published:** 2025-12-16

**Authors:** Patrycja Głowacka, Izabela Marczuk, Patrycja Wójcicka, Monika Ogórkiewicz, Marta Ciesielska, Dorota Żakowska, Paweł Rutyna, Anna Koszczyńska, Marta Łączyńska, Natalia Podsiadły, Emilia Paziewska, Beata Cieśluk-Olchowska

**Affiliations:** 1Biological Threats Identification and Countermeasure Centre of the Military, Institute of Hygiene and Epidemiology, 24-100 Puławy, Poland; izabela.marczuk@wihe.pl (I.M.); marta.ciesielska@wihe.pl (M.C.); dorota.zakowska@wihe.pl (D.Ż.); 2Chair and Department of Medical Microbiology, Medical University, 20-059 Lublin, Poland; patrycja.wojcicka@umlub.edu.pl (P.W.); pawel.rutyna@umlub.edu.pl (P.R.); 3Medical Laboratory Department, 10th Military Clinical Hospital with Polyclinic, 85-681 Bydgoszcz, Poland; 4Microbiological Diagnostics Laboratory, 105th Borderland Military Hospital with an Outpatient Clinic, 68-200 Żary, Poland; n.podsiadly@105szpital.pl; 5Laboratory Medicine Department, 109th Military Hospital with an Outpatient Clinic, 71-422 Szczecin, Poland; 6Medical Laboratory Department, 1st Military Clinical Hospital with Policlinic in Lublin, Branch, 19-300 Ełk, Poland

**Keywords:** carbapenems, genes, β-lactams

## Abstract

**Background**: There are many methods of identifying microbial resistance to therapeutic agents; however, they can generally be classified into two main categories: phenotypic and genotypic. The study aims to determine drug sensitivity and to analyze the correlation between the results obtained from cultures on commercial chromogenic media Brilliance^TM^ CRE (OXOID) and Brilliance^TM^ ESBL (OXOID) and the occurrence of specific resistance genes carbapenemase (IMP, NDM, VIM, KPC, OXA), ESBL β-lactamase (TEM, SHV, CTX-M), and AmpC (CMY, DHA), which will be used in drug sensitivity tests. **Methods**: The present study used bacteria, including *Klebsiella pneumoniae*, *Acinetobacter baumannii*, and *Escherichia coli*, obtained from patients hospitalized in military hospitals in Poland. All strains were plated on the commercial chromogenic media and subjected to antimicrobial susceptibility testing. Additionally, molecular assays detecting three main classes according to the mechanism of action, enzyme type carbapenemase (IMP, NDM, VIM, KPC, OXA), ESBL β-lactamase (TEM, SHV, CTX-M), and AmpC (CMY, DHA) were performed using the real-time PCR method. **Results**: The results of the studies indicate the presence of carbapenemases and ESBL genes. Among *K. pneumoniae* strains, the dominant gene was CTX-M-15 (88.89%), followed by the SHV (84.12%), NDM (46.03%), TEM (41.26%), KPC (34.92%), and OXA-48 (19.04%). In contrast, *A. baumanii* was dominated by carbapenemases from the OXA family (OXA-51 in 96.00% and OXA-24/40 in 84.00%). *E. coli* exhibits a high prevalence of CTX-M-15 (53.85%), TEM (46.15%), NDM (38.46%), and CMY-2 (30.77%). It was observed that the CTX-M-15 gene was commonly co-identified with SHV (*n* = 43). All tested strains grew on chromogenic Brilliance^TM^ CRE medium. In the case of Brilliance^TM^ ESBL medium, the genes determining the resistance mechanism were detected in 41.7% for *A. baumannii*, 53.8% for *E. coli*, and 100% for *K. pneumoniae*. Chromogenic media perfectly differentiate strains to species. A moderate positive correlation of the occurrence of the antibiotic resistance genes was observed for OXA-51 and OXA-24/40 genes, which were resistant to meropenem (rho = 0.45, *p* < 0.001). K-means cluster analysis performed on integrated genotype–phenotype data allowed for the identification of three distinct clusters characterized by distinct resistance gene profiles. These results demonstrate that selective agar media enable faster identification compared to other conventional techniques; however, the obtained results should be confirmed by other validated phenotypic methods, and, if possible, by a molecular assay.

## 1. Introduction

Antimicrobial resistance (AMR) is currently a global issue of the 21st century. The resistance of microorganisms to antibacterial substances is now one of the most significant epidemiological challenges on a global scale. It is estimated that bacteria resistant to commonly used antibiotics kill up to 700.000 people worldwide every year. The spread of multidrug-resistant strains is driven by the widespread use of antibiotics in medicine, veterinary practice, and agriculture. Additionally, the ease with which genes are transmitted between bacteria significantly limits therapeutic options for many bacterial infections. It is important to highlight the role of commensal strains in the dissemination of antibiotic resistance. Such bacteria can serve both as a reservoir and a vector of genes responsible for antimicrobial resistance, thus posing a threat especially to individuals with weakened immune systems [1]. Bacteria, like other organisms, according to Darwin’s theory, undergo a process of evolution in order to adapt to the prevailing environmental conditions and to escape from substances that cause their death. Some of the microorganisms evolved mechanisms that protect them against the drug’s lethal effect. They developed various ways to avoid antibiotics, such as changes in the target site, repression in penetration and distribution, or production of enzymes that break down antibiotics [2]. At present, the most significant concern is the resistance of bacteria associated with high mortality rates, such as *Klebsiella pneumoniae*, *Pseudomonas aeruginosa*, and *Acinetobacter baumannii* [1]. According to a document published by the WHO in 2024, the pathogens posing the greatest threats to public health include: carbapenem-resistant Enterobacterales, third-generation cephalosporin-resistant Enterobacterales, and carbapenem-resistant *Acinetobacter baumannii* [3].

There are many methods for identifying microbial resistance to therapeutic agents; however, they can generally be classified into two main categories: phenotypic methods and genotypic methods [4].

Among phenotypic methods, one can distinguish classical antibiotic susceptibility tests such as the disk diffusion method, gradient diffusion, and broth microdilution, as well as CIM (Carbapenem Inactivation Method) and CarbaNP tests. An important component is the availability of commercial automated systems (e.g., Phoenix BD [Becton Dickinson, USA], Vitek 2 [bioMérieux, France]), which, in combination with compatible tests, enable the identification and antimicrobial susceptibility testing of microorganisms [5].

The availability and wide selection of selective chromogenic media is another factor that has enabled the preliminary detection and identification of resistant bacterial strains based on colony color. Currently available chromogenic media can identify resistance to antibiotics such as carbapenems (including OXA-type), extended-spectrum beta-lactamases (ESBLs), vancomycin (VRE), methicillin (MRSA), colistin, and linezolid. Culturing samples on these media can be performed directly from clinical specimens, which also contributes to the rapid obtaining of results that may guide further actions. The growth of characteristic colonies on a given medium, in most cases, indicates the cultivation of a strain with the corresponding resistance mechanism. However, it is important to note that, especially in the case of media with a broad spectrum of detectable mechanisms, such as those used for carbapenemase or ESBL identification, growth of a strain may result from the presence of other mechanisms, e.g., ESBL or AmpC overexpression (in the case of carbapenemases) or altered permeability of cell membranes. Therefore, to prevent false-positive or false-negative results, interpretations should be made cautiously, taking into account antimicrobial susceptibility results and other diagnostic tests [5,6].

Recently, molecular methods have emerged as an essential addition to phenotypic approaches. These techniques are primarily based on conventional PCR (polymerase chain reaction) and real-time PCR (RT-PCR) [7]. Ready-to-use commercial kits with IVD (In Vitro Diagnostics) certification are available on the market for identifying the most commonly occurring resistance genes. Additionally, there is the option to use a wider range of available kits labeled as RUO (Research Use Only), as well as the possibility to design primers and reaction conditions independently for targeting specific genes, although this is limited to research purposes only [8]. Due to the serious threat posed by carbapenem-resistant strains, there is particular interest in searching for additional genes encoding this resistance. In the case of carbapenemases, molecular diagnostics largely rely on detecting genes from representatives belonging to three classes: A, B, and D (according to molecular differentiation based on Ambler classification). The first class, class A, also known as serine carbapenemases, includes the KPC gene (*K. pneumoniae* carbapenemase) [9]. This is one of the most widespread carbapenemases, mainly found in Enterobacterales, and strongly hydrolyzes all β-lactams, including carbapenems. Class B genes belong to MBLs (metallo-β-lactamases dependent on zinc ions), which include NDM (New Delhi metallo-β-lactamase), widely distributed and conferring resistance to all β-lactams except monobactams (e.g., aztreonam), VIM (Verona integron-encoded metallo-β-lactamase) found in *P. aeruginosa*, Enterobacterales, and hydrolyzes carbapenems, and IMP (Imipenem metallo-β-lactamase), present in *P. aeruginosa* and *A. baumannii*, acting similarly to VIM [1,5]. In class D, similar to class A, there are serine β-lactamases with carbapenemase activity, specifically carbapenemases from the OXA group. The most common include OXA-48, found in Enterobacterales—hydrolyzes carbapenems—has weak activity against cephalosporins, and is difficult to detect in routine susceptibility tests, OXA-51, naturally present in *A. baumannii*, which can confer carbapenem resistance, especially after increased expression, and OXA-23, common in *A. baumannii*, strongly hydrolyzes carbapenems but has no activity against cephalosporins. OXA-58, also found in *Acinetobacter*, has reduced activity against carbapenems compared to OXA-23. OXA-143 is a less common carbapenemase in *Acinetobacter*, similar to OXA-23. OXA-24/40, present in *A. baumannii*, confers high-level carbapenem resistance. OXA-482, a variant of OXA-48, was detected in Enterobacterales, resistant to carbapenems [10,11]. In the literature, the above group is often referred to as the “big five,” as the mentioned genes encoding enzymes are responsible for the most commonly occurring carbapenemases worldwide [12,13,14]. A similar situation applies to the identification of genes determining the mechanism of resistance of the ESBL type, which is based on the expression of genes encoding enzymes that hydrolyze the β-lactam ring. Genes responsible for this type of resistance are most often located on mobile genetic elements such as plasmids or transposons [1,5,9]. According to the Ambler classification of β-lactamases, ESBLs are assigned to classes A and D [15]. A characteristic feature of these groups is the presence of serine in the active site [16]. The most frequently detected ESBL resistance genes belong to class A and include SHV (sulfhydryl reagent variable β-lactamases)—a broad-spectrum β-lactamase often found in *K. pneumoniae*, TEM (Temoniera β-lactamases)—one of the most common β-lactamases, occurring in variants such as TEM-1 and TEM-2 (classic β-lactamases hydrolyzing penicillins and aminopenicillins), and TEM-3 and higher variants that hydrolyze third-generation cephalosporins, CTX-M (cefotaxime-M β-lactamases)—with an extended spectrum of activity against cephalosporins, including types such as CTX-M-1, CTX-M-2, CTX-M-8, CTX-M-9, CTX-M-14, CTX-M-15, and CTX-M-25 [15,17]. Equally frequently observed is the presence of the GES (Guiana extended-spectrum) gene—a broad-spectrum β-lactamase, some variants of which (e.g., GES-5) may also exhibit carbapenemase activity [18]. A separate class C comprises AmpC-type β-lactamases, located both on plasmid DNA and chromosomal DNA [19]. They confer resistance to cephalosporins (III–IV generation), penicillins, and sometimes carbapenems. Currently, the literature distinguishes the following AmpC clusters: CMY-2 found in *Escherichia coli*, *Proteus mirabilis*, and *K. pneumoniae*, which hydrolyze cephalosporins, DHA, originally identified in *K. pneumoniae* and also present in *P. mirabilis* [20].

Although gene detection using molecular techniques is a precise tool with high sensitivity and specificity, this method also has its limitations. The most important of these include: the possible lack of expression of a given gene, the use of primers targeted only to amplify specific genes according to selected targets, as well as the emergence of new variants of a given gene whose original primer sequence may no longer match. Incorrectly designed primers, insufficient diversity, or insufficient quantity of primers for a given reaction may result in false-negative results. Therefore, results obtained by PCR and RT-PCR methods should be correlated with phenotypic testing [7].

## 2. Results

### 2.1. Growth on Chromogenic Media Brilliance™ CRE and Brilliance™ ESBL

According to information provided by the manufacturer, Brilliance™ CRE agar (OXOID) allows for the identification of Enterobacterales (*E. coli*, *Klebsiella* sp., *Citrobacter* sp., *Serratia* sp., *Enterobacter* sp.) as well as *Acinetobacter* sp.-producing enzymes known as carbapenemases. Brilliance™ ESBL agar (OXOID), due to its chromogenic substances and the addition of cefpodoxime, inhibits the growth of *Enterobacteriaceae* that do not produce beta-lactamases and enables differentiation of bacteria from the KESC group (*Klebsiella* sp., *Enterobacter* sp., *Serratia* sp., *Citrobacter* sp.) from *E. coli.* Additionally, this medium differentiates *E. coli* strains based on the production of galactosidase and glucuronidase. In the presented study, all tested isolates grew on the chromogenic medium Brilliance™ CRE (OXOID) (described in Figure 1). Twelve *A. baumannii* isolates grew on the chromogenic medium Brilliance™ ESBL (OXOID) despite the absence of detected ESBL resistance genes. Five isolates growing on Brilliance™ ESBL (OXOID) carried TEM resistance genes. Among the *E. coli* isolates that grew on Brilliance™ ESBL (OXOID), no resistance genes associated with the tested mechanism were found in 6 isolates, while 7 isolates possessed resistance genes conferring the ESBL mechanism, such as CTX-M-15 and TEM. Sixty *K. pneumoniae* isolates that grew on Brilliance™ ESBL (OXOID) harbored ESBL resistance genes. The above results is presented in Figure 2. The positive predictive value (PPV), which indicates how often growth on the medium corresponds to the presence of resistance genes, was 41.7% (5/12) for *A. baumannii*, 53.8% (7/13) for *E. coli*, and 100% (60/60) for *K. pneumoniae*. False positive growth, growth without antibiotic resistance genes, occurred in 58.3% (7/12) of *A. baumannii* and 46.2% (6/13) of *E. coli* isolates. These false positive isolates grew as described by the manufacturer, as shown in the images below. Among the *E. coli* strains, 1 out of 13 was sensitive to ceftazidime and cefepime, while the remaining isolates were resistant to ceftazidime, cefepime, and cefotaxime. Among the *K. pneumoniae* isolates, one was sensitive to ceftazidime and cefotaxime, another was sensitive to cefotaxime alone, and the rest showed resistance to ceftazidime, cefepime, and cefotaxime. Seven *K. pneumoniae* strains had phenotypically confirmed ESBL resistance mechanisms.

The Brilliance™ CRE medium, according to the manufacturer’s instructions, is designed for the identification of carbapenem-resistant bacteria. All tested bacterial strains (*K. pneumoniae*, *A. baumannii*, *E. coli*) in which resistance genes to carbapenems of groups A, B, and D, as well as ESBL and cephalosporin resistance (detection of *AmpC* group genes), were detected, exhibited a specific growth pattern with colony morphology characteristic for each species. The use of the chromogenic Brilliance™ CRE medium enables the cultivation of microorganisms producing various classes of beta-lactamases; however, caution should be exercised in clinical interpretation due to the occurrence of false positive results, which suggests the need to use methods to confirm the obtained results with other methods. The Brilliance™ ESBL medium promotes the growth of specific bacterial species in the form of colonies with characteristic coloration. For species exhibiting the ESBL resistance mechanism (in this study, detection of CTX-M-14, CTX-M-15, SHV, TEM genes), the colony growth pattern corresponded to the manufacturer’s specifications. However, the study observed that growth occurred for all tested species regardless of the presence of the ESBL resistance mechanism. Strains lacking this mechanism still grew as colonies colored according to the manufacturer’s guidelines. The genetic profile of strains producing false positive results on ESBL medium was OXA-24/40 and OXA-51. For *E. coli*, the genetic profile of false positive samples involved the CMY and NDM genes. No false positive results were observed for *K. pneumoniae*. This medium is most effective in detecting *K. pneumoniae*, for other species, growth may also occur in strains producing carbapenemases. Therefore, this medium is not useful for determining the presence or absence of the ESBL resistance mechanism. Nevertheless, it correctly differentiates bacterial species by producing colonies with distinct coloration.

### 2.2. Antimicrobial Susceptibility Assessment Using Phenotypic Methods

All *A. baumannii* strains exhibit phenotypic resistance to imipenem, ciprofloxacin, and trimethoprim/sulfamethoxazole. One isolate tested indicated increased sensitivity (described in Figure 3, Figure 4 and Figure 5 as “S”) to meropenem, while others were resistant to this antibiotic. A similar situation occurred with levofloxacin. One isolate was sensitive to amikacin, while the others were resistant (described in Figure 3, Figure 4 and Figure 5 as “R”). In terms of gentamycin, six isolates were susceptible, two were susceptible with increased exposure (described in Figure 3, Figure 4 and Figure 5 as “I”), and seventeen were resistant. All of the *E. coli* isolates tested were resistant to ampicillin, amoxicillin/ clavulanic acid, piperacillin, and cefotaxime. Seven of them were susceptible to gentamycin, piperacillin, cefoxitin, imipenem, meropenem, and amikacin. Only one of the *E. coli* strains was resistant to nitrofurantoin. All *K. pneumoniae* isolates were resistant to ampicillin, amoxicillin/clavulanic acid, piperacillin, and cefepime. Susceptibility to ertapenem was indicated in one isolate, two isolates remained sensitive to cefotaxime, and three isolates were sensitive to ertapenem. The summary of antimicrobial resistance among *A. baumannii*, *E. coli*, and *K. pneumoniae* isolates is presented in Figure 3, Figure 4 and Figure 5.

### 2.3. Molecular Analysis of the Occurrence of Resistance Genes

Among *K. pneumoniae*, the following genes were detected: NDM (*n* = 29, 46%), KPC (*n* = 22, 35%), OXA-48 (*n* = 12, 19%), TEM (*n* = 26, 41%), SHV (*n* = 53, 84%), CTX-M-15 (*n* = 56, 89%). In *A. baumannii* isolates detected: OXA-24/40 (*n* = 21, 84%), OXA-51 (*n* = 24, 96%), OXA-23 (*n* = 3, 12%), TEM (*n* = 5, 20%), while in *E. coli* isolates identified NDM (*n* = 5, 38%), VIM (*n* = 1, 7%), KPC (*n* = 1, 7%), TEM (*n* = 6, 46%), CTX-M-15 (*n* = 7, 53%), CMY-2 (*n* = 4, 30%), DHA (*n* = 1, 7%). The assay of the isolates revealed no presence of the CTX-M-14, OXA-58, or OXA-143 genes. CTX-M-15 was the most frequently found resistance gene, present in 57.8% of the tested samples. The above data is presented in summary in Table 1.

The CTX-M-15 gene was co-identified with SHV (*n* = 43), NDM (*n* = 26), TEM (*n* = 21), KPC (*n* = 20), GES (*n* = 6), and OXA-48 (*n* = 7). The second most common resistance gene was SHV, which occurred with the genes NDM (*n* = 23), KPC (*n* = 17), TEM (*n* = 18), OXA-48 (*n* = 6), OXA-51 (*n* = 3), and GES (*n* = 5). The third most common resistance gene was NDM, which occurred together with CTX-M-15 (*n* = 26), SHV (*n* = 23), OXA-48 (*n* = 5), CMY-2 (*n* = 4), TEM (*n* = 4), and GES (*n* = 2). The gene co-occurrence matrix is illustrated below in Figure 6. A high frequency of co-occurrence of genes OXA-51 and OXA-24/40 was observed. An important rule of gene co-occurrence has also been observed among genes CTX-M-15, TEM, SHV, and KPC.

### 2.4. Correlation of the Occurrence of Resistance Genes with Antibiotic Resistance

A moderate positive correlation was observed for the OXA-51 and OXA-24/40 genes and resistance to meropenem (rho = 0.45, *p* < 0.001). The presence of the following genes might be associated with increased resistance to meropenem and, to a lesser extent, resistance to imipenem (rho = 0.29; *p* = 0.0061). The presence of the NDM gene correlates with higher resistance to imipenem (rho = 0.29; *p* = 0.0071) and ertapenem (rho = 0.32; *p* < 0.001); however, it is not a strong correlation. A correlation between the presence of the NDM gene and resistance to cefoxitin (rho = 0.44; *p* < 0.001), piperacillin/ tazobactam (rho = 0.32; *p* < 0.001), and in a lesser extent, resistance to cefotaxime (rho = 0.15; *p* < 0.001), ceftazidime (rho = 0.15; *p* < 0.001), and cefepime (rho = 0.1; *p* = 0.13) has also been observed. The occurrence of the CTX-M-15 gene may be connected with resistance to cefotaxime (rho = 0.31, *p* < 0.001), ceftazidime (rho = 0.31, *p* < 0.001), and cefepime (rho = 0.19, *p* = 0.0004). The SHV gene has a weak correlation with the occurrence of resistance to piperacillin/ tazobactam (rho = 0.26, *p* < 0.001), cefotaxime (rho = 0.21, *p* < 0.001), ceftazidime (rho = 0.21, *p* < 0.001), cefepime (rho = 0.19, *p* < 0.001), and to a lesser extent with resistance to cefoxitin (rho = 0.11, *p* < 0.001). Spearman’s rank correlation coefficients and *p*-values for all genes studied and phenotypic antibiotic resistance are shown in Figure 7 and Figure 8.

### 2.5. PCA of Antimicrobial Resistance and Genotypic Profiles

The use of the K-means algorithm allowed for the identification of the three clusters with diverse genetic and phenotypic profiles, as visualized in Figure 9. The percentage of species in cluster 0 is 86% *K. pneumoniae* and 14% *E. coli*, isolates in which ESBL resistance genes were detected, but no carbapenemases were present. Cluster 1 is composed of 76% *K. pneumoniae*, 21% *E. coli* and 3% *A. baumanii* with carbapenemases present and resistance to beta-lactam antibiotics. Cluster 2 consists almost exclusively of *A. baumannii* isolates in which OXA-23, OXA-24/40, and OXA-51 genes were detected. This separation results from differences in resistance gene profiles and antimicrobial resistance patterns, which were variables in the PCA and k-Mears analyses. The distinct cluster formed by *A. baumanii* results from its characteristic distinct genetic profile formed by OXA carbapenemase genes. K-means cluster analysis performed on integrated genotype–phenotype data revealed the presence of three distinct resistance profiles, characterized by different combinations of carbapenemase and beta-lactamase genes and varying levels of antibiotic resistance. Cluster 0 is characterized by a moderate presence of ESBL genes, primarily CTX-M-15 (62%), and a low prevalence of class A (KPC 19%) and class D (OXA-48 6%) carbapenemases. Phenotypically, these strains exhibited low levels of aminoglycoside resistance (AK: 12%) and lower levels of carbapenem resistance (MEM: 0%, IMP: 6%) compared to the other clusters. This cluster exhibited high levels of cephalosporin resistance (FEP: 94%, CAZ: 94%). Cluster 1 is characterized by the highest prevalence of metallo-beta-lactamases: NDM (58%), and class A and D carbapenemases, i.e., KPC (30%) and OXA-48 (17%). Furthermore, there is a high prevalence of the ESBL gene CTX-M-15 (82%). The presence of AmpC CMY-2 (12%), GES (55%), and TEM (55%) has also been observed. Strains grouped in this cluster are characterized by the highest levels of resistance to carbapenems (MEM: 57%, IMP: 97%), and broad resistance to aminoglycosides (AK: 48%), fluoroquinolones, and cephalosporins (FEP: 76%, CAZ: 83%). These strains have an MDR profile. These strains constitute the most clinically problematic group with limited therapeutic options. Cluster 2 is the cluster with the dominant OXA-51 (88%) and OXA-24/40 (77%) genes. A low presence of class A carbapenemase genes (KPC 4%) was observed, and ESBL, AmpC, and metallo-β-lactamases were absent in this cluster. These strains phenotypically exhibited high-level resistance to carbapenems (MEM: 88%, IMP: 92%) and aminoglycosides (AK: 88%), and low-level resistance to cephalosporins (FEP: 11%, CZA: 12%). Resistance to quinolones was high in all clusters: cluster 0 (81%), cluster 1 (88%), cluster 2 (88%).

## 3. Discussion

Antibiotic resistance is a major public health challenge, exacerbated by the widespread use of β-lactam and glycopeptide antibiotics. In the analyzed materials, a total of 101 isolates of Gram-negative bacteria were tested, of which the isolated genes representing the ESBL resistance mechanism were CTX-M-15 (56), SHV (53), and TEM (26) in *K. pneumoniae*; TEM (6) and CTX-M-15 (7) in *E. coli* and TEM (5) in *A. baumanii*. The highest positive predictive value (PPV), indicating how often growth on the medium indicates the presence of the resistance gene, was 100% (63/63) for *K. pneumoniae*, while for *A. baumanii* it was 41.7% (5/12) and *E. coli* 53.8% (7/13). In further work, studies using a wider panel of chromogenic substrates are planned [21]. In a subsequent study, the aim of the study by Te-Din Huang et al. was to evaluate the efficacy of another chromogenic medium, Brilliance ESBL agar (OX; Oxoid, Basingstoke, UK) and ChromID ESBL (BM; bioMérieux, Marcy l’Etoile, France), for the selective isolation and presumptive identification of extended-spectrum beta-lactamase (ESBL)-producing Enterobacteriaceae. A panel of 200 clinical isolates of Gram-negative and nonfermentative Enterobacteriaceae with defined resistance mechanisms was inoculated onto chromogenic OX medium and ChromID ESBL agar (BM; bioMérieux, Marcy l’Etoile, France). Of the 156 Enterobacteriaceae isolates, 8 fully susceptible isolates were inhibited, all 98 ESBL producers were detected, and 50 isolates exhibiting other resistance mechanisms were isolated on both chromogenic agars. In the second phase, 528 clinical samples (including 344 stool samples) were plated on OX, BM, and MacConkey agar with a ceftazidime disk (MCC) to screen for ESBL-producing Enterobacteriaceae. Growth on at least one medium was observed in 144 (27%) of the clinical samples tested. A total of 182 isolates were isolated, including 109 (60%) Enterobacteriaceae, and 70 of these (from 59 samples) were confirmed as ESBL-producing isolates. The sensitivities of MCC, BM, and OX were 74.6%, 94.9%, and 94.9%, respectively. The specificities of MCC, BM, and OX samples reached 94.9%, 95.5%, and 95.7%, respectively, when only colored colonies were included on the two selective chromogenic media. The high positive predictive value (99.3%) for OX indicates that this medium may be an excellent screening tool for rapidly excluding patients who do not carry ESBLs [22]. The ability of carbapenemases to degrade all β-lactam antibiotics leads to fewer antibiotics retaining activity against infections caused by CPE (Carbapenemase-producing Enterobacteriaceae), which are associated with high mortality and poor prognosis. The use of Chromatic™ CPE medium for the identification of CRE (Carbapenem-resistant Enterobacteriaceae) directly from ICU (Intensive Care Unit) clinical samples has demonstrated high efficacy. One hundred eight isolates were obtained from patients admitted to the treatment unit at Ain Shams University Hospital between April and November 2022. There are various isolates, but the most common are sputum-derived. The sensitivity and specificity of the CRE screening test for chromogenic CRE, accounting for carbapenemase inactivation (mCIM) as standard. The sensitivity of the chromogenic medium was 98%, the specificity was 50% for all isolates, and the accuracy of the test was 94.4%. The sensitivity of the chromogenic medium for all Enterobacteriaceae was 96.9%, and the specificity was 33.3%. The study demonstrated that chromogenic CRE medium is highly sensitive for CRE screening. use of a single chromogenic medium can reduce sample processing costs and be an effective tool for rapid CRE detection. Accurate and rapid detection in patients colonized with CPE is clinically important for implementing appropriate infection control measures [23]. In a conducted study aforementioned high sensitivity and specificity has been partially confirmed. These observations apply primarily to *K. pneumoniae*, where the result obtained on the chromogenic medium was fully reflected in the identified genes. However, significantly lower results were obtained for the other strains (*A.baumanii*, *E.coli*) It should also be mentioned that chromogenic media allowed for the faultless initial identification of the tested strains. The genotypic results obtained in the conducted studies, determining the presence of carbapenemases and ESBL enzymes in *K. pneumoniae*, *A. baumanii*, *and E. coli* isolates, emphasize the heterogeneity of resistance mechanisms in these species. The obtained results are consistent with previous reports emphasizing the production of carbapenemases by this species. In a study conducted in 2016 in Gabon, the most common CP-GNB (Carbapenemase-Producing Gram-Negative Bacteria) were *K. pneumoniae* (53.33%) and *A. baumannii* (26.67%), *E.coli* (13.33%) which is comparable with our study. BlaOXA-48 was the dominant carbapenemase gene (40%), followed by blaNDM-5 (33.33%), while in our study, the most popular carbapenemase genes were, respectively, NDM (33%), KPC (22%), and OXA-48 (12%). The reported results indicate a very similar prevalence of NDM genes, whereas the occurrence of the OXA-48 gene is significantly lower in our study. Interestingly, studies conducted in Gabon did not detect the KPC gene, while in our study, it accounted for as much as 22% of the carbapenem resistance-associated genes [24]. Among *K. pneumoniae* strains, the dominant gene is NDM (46.03%), followed by the occurrence of KPC (34.92%) and OXA-48 (19.04%). In contrast, *A. baumanii* is dominated by carbapenemases from the OXA family (OXA-51 in 96% and OXA-24/40 in 84%). The absence of NDM, VIM, or KPC genes was observed, indicating a significant role of OXA enzymes in the resistance of this pathogen. Previous reports emphasize the dominant role of OXA-51 and OXA-24/40 in *A. baumanii*. *A. baumanii* isolates carry almost exclusively OXA enzymes, without NDM, VIM, or KPC. In a study conducted in Nepal on 382 *A. baumanii* samples, similar levels of OXA gene occurrence were achieved to those obtained in our results: OXA-51 was present in 96% of isolates, while OXA-24/40 was present in 84% of isolates [25]. In this study, *E. coli* exhibits a high prevalence of CTX-M-15 (53%), TEM (46%), and CMY-2 (30%), but also the presence of NDM (38%), and a low prevalence of VIM (7%) and KPC (7%) was observed. These results demonstrate the diversity of resistance mechanisms, particularly with respect to ESBL enzymes. All three species exhibit diverse resistance profiles, indicating the need for further research and epidemiological monitoring, which will directly translate into more effective antibiotic therapy and limit the spread of multidrug-resistant strains. Published studies indicate a high prevalence of CTX-M genes, ranging from 21 to 96%, with some authors also indicating the CTX-M-1 (53%) and CTX-M-8 (24%) subtypes. The presence of the TEM gene was reported in 16%, 58%, 59%, and 72% of the *E. coli* strains tested. The abundance of SHV genes in *E. coli* was determined at levels of 16% and 32%. The CTX-M-15 gene is the most frequently isolated EBSL gene [25,26,27]. In summary, resistance detection is crucial for infection control policies, and the availability of selective agar media allows for faster identification of infected patients compared to other conventional techniques and allows for the immediate implementation of infection control measures to prevent further spread, eliminating the need for inappropriate antibiotic therapy [28]. However, these methods also have some drawbacks. It is widely known that phenotypic methods have significant limitations. For example, depending on their environment microorganisms can alter their gene expression. In such situations, genetically identical microorganisms may display different phenotypic traits. Point mutations may also occur in bacterial genomes, leading to changes in phenotype. Therefore, phenotypic and genotypic methods complement each other and should be used simultaneously in order to achieve the best possible understanding of the mechanisms of antibiotic resistance [4].

## 4. Materials and Methods

### 4.1. Bacterial Strains

Bacterial strains (*n* = 101) used in the following study were collected from patients hospitalized in 5 military hospitals located in Poland. Primary isolate material obtained: blood culture (*n* = 38), rectal swabs (*n* = 36), bronchoalveolar lavage BAL (*n* = 15), urine culture (*n* = 5), PJC (*n* = 1), wound swab (*n* = 3), pressure sore swab (*n* = 1), blood from the injection (*n* = 1), a swab from a leg ulcer (*n* = 1). All samples were identified to species in hospital microbiology laboratories and selected purified bacterial isolates submitted as pure strain isolates on transport medium to the Biological Threats Identification and Countermeasure Center in Puławy for further molecular analysis. Among the samples sent for testing, 63 isolates belonged to *Klebsiella pneumoniae*, 25 isolates to *Acinetobacter baumannii*, and 13 isolates to *Escherichia coli*. The number of microorganisms by origin of the original sample is presented in Table 2. Acquired isolates were cultured on blood agar plates supplemented with 5% sheep blood (Columbia agar, Biomaxima, Poland). Plates were incubated at 37 °C for 24 h.

#### 4.1.1. Chromogenic Media

After 24 h of incubation on blood agar, all samples were plated on the following commercial chromogenic media Brilliance^TM^ CRE (OXOID, Basingstoke, UK) and Brilliance ^TM^ ESBL (OXOID, Basingstoke, UK). Plates were incubated at 37 °C for 24 h. After incubation, the plates were read according to the manufacturer’s guidelines, as follows:

Brilliance^TM^ CRE—all strains growing as blue colonies are KESC (*Klebsiella* sp., *Enterobacter* sp., *Serratia* sp., *Citrobacter* sp.) carbapenem-resistant bacteria; bacteria growing as white or naturally pigmented colonies are carbapenem-resistant *Acinetobacter* sp.

Brilliance^TM^ ESBL—green colonies belonged to the ESBL-resistance KESC group; blue colonies are ESBL-resistant *E.coli* (galactosidase-positive); pink colonies are ESBL-resistant *E. coli* (galactosidase-negative); light-brown colonies—ESBL-resistant *Proteus* sp., *Morganella* sp., *Providencia* sp.; colourless colonies—ESBL-resistant *Salmonella* sp., *Acinetobacter* sp.

As a positive control, the study used the following strains: *K. pneumoniae* ATCC 700603—ESBL positive, *K. pneumoniae* NCTC 13438—KPC positive; *K. pneumoniae* NCTC 13442-OXA-48 positive. As a negative control, susceptible *K. pneumoniae* ATCC 25955 was applied.

#### 4.1.2. Antimicrobial Susceptibility Testing—Disc Diffusion Methodology

Kirby–Bauer disk diffusion method in accordance with the European Committee on Antimicrobial Susceptibility Testing v 15.0 (EUCAST) guidelines was used to assess the phenotypic drug resistance of the tested strains. For this purpose, 0.5 McFarland inoculum of each 24-h strain culture was prepared and cultured on Mueller–Hinton agar (Biomaxima, Poland). Discs containing antibiotics were then applied (OXOID, Basingstoke, UK). The plates were incubated at 35 ± 1 °C for 18 ± 2 h. The antimicrobial agents selected for the study for the tested bacterial species are presented in Table 3. The inhibition zones were measured and interpreted according to the EUCAST guidelines (Version 15.0, valid from 1 January 2025) [29].

### 4.2. Molecular Methods RT-PCR Assay

The total DNA from bacterial colonies was isolated using the commercial SaMag Bacterial DNA Extraction Kit (SaMag-12, Sacace, Como CO, Italy). Molecular assays were performed using the real-time PCR method with the commercial Streck ARM-D Kit (Streck, La Vista, NE, USA). The following tests were used during the analyses: Streck ARM-D Kit β-Laktamase, identifying resistance genes from the family of KPC, ESBL, MBL; Streck ARM-D Kit OXA, identifying OXA-like genes; Streck ARM-D Kit TEM/SHV/GES, identifying resistance genes from the ESBL family; Streck ARM-D Kit ampC, identifying cephalosporin resistance genes. All particular genes from the tests are listed in Table 4. In order to verify the correctness of the reaction, a positive control provided by the manufacturer was used.

### 4.3. Software and Statistical Analysis

The following libraries were used to create the charts: Pandas version 2.2.3, Matplotlib version 3.10.0, and Seaborn version 0.13.2.

## Figures and Tables

**Figure 1 antibiotics-14-01275-f001:**
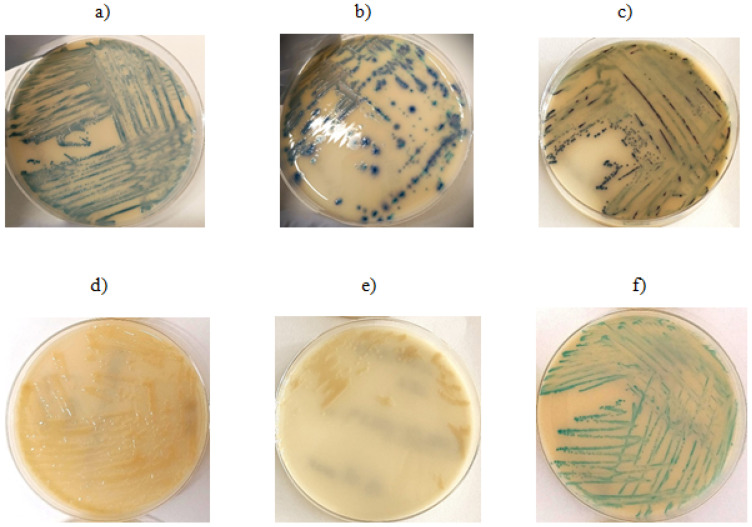
Growth on ESBL chromogenic medium: (**a**) *E. coli* isolate with detected ESBL resistance gene, (**b**,**c**) *E. coli* isolates—no resistance gene detected, (**d**,**e**) *A. baumannii* isolates without detected resistance genes, (**f**) *K. pneumoniae* isolate with detected resistance gene.

**Figure 2 antibiotics-14-01275-f002:**
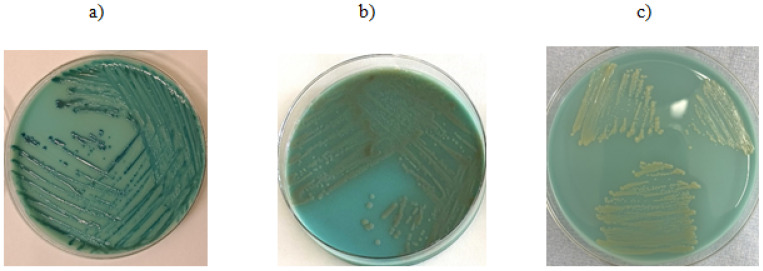
Growth on CRE chromogenic medium: (**a**) *K. pneumoniae*—isolate shows no presence of the resistance mechanism gene, (**b**) *E. coli*—isolate shows no presence of the resistance mechanism gene, (**c**) *A. baumannii*—PCR did not detect the presence of the resistance mechanism gene.

**Figure 3 antibiotics-14-01275-f003:**
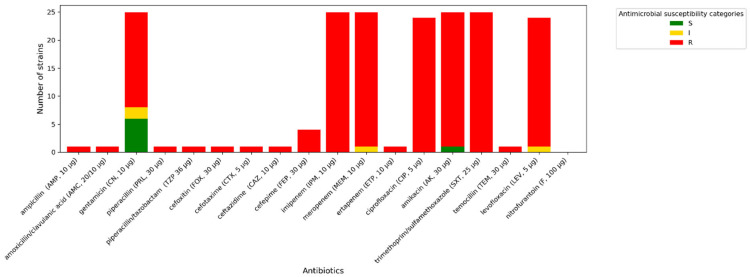
Amount of antimicrobial-resistant *A. baumannii* strains.

**Figure 4 antibiotics-14-01275-f004:**
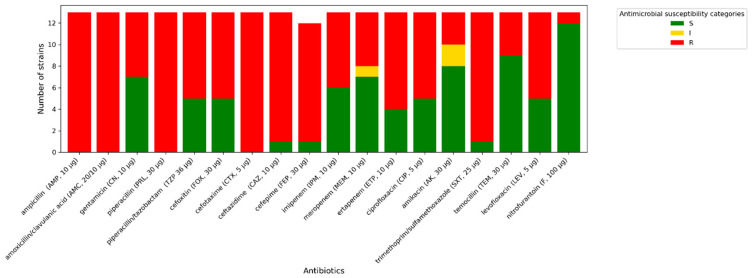
Amount of antimicrobial-resistant *E. coli* strains.

**Figure 5 antibiotics-14-01275-f005:**
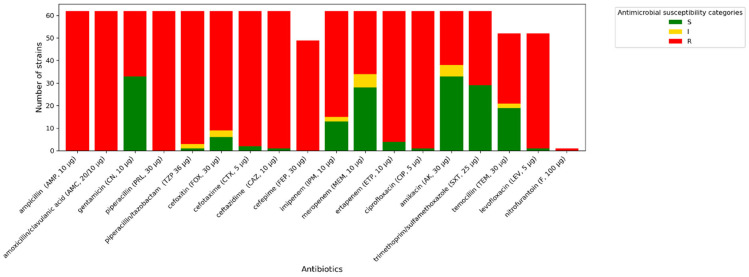
Amount of antimicrobial-resistant *K. pneumoniae* strains.

**Figure 6 antibiotics-14-01275-f006:**
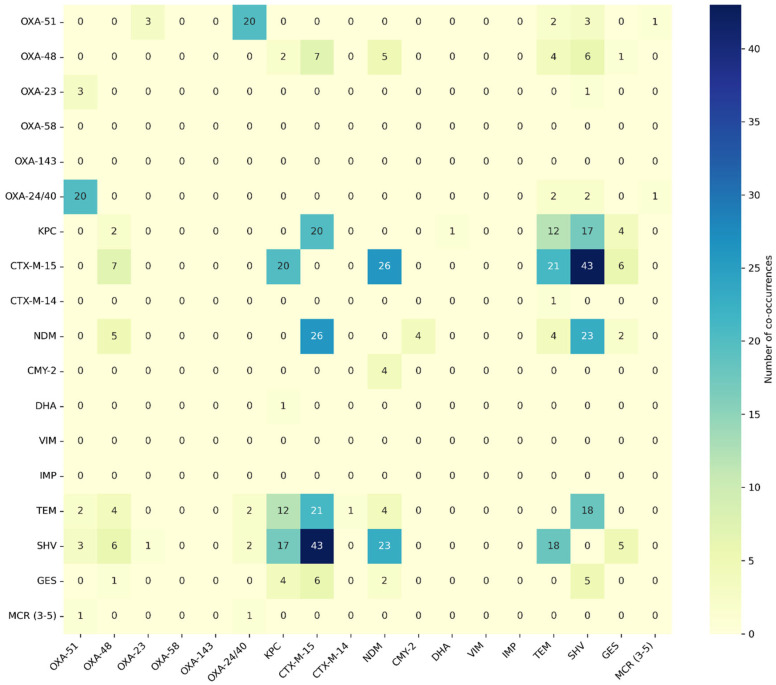
Heat map of resistance genes co-occurrence.

**Figure 7 antibiotics-14-01275-f007:**
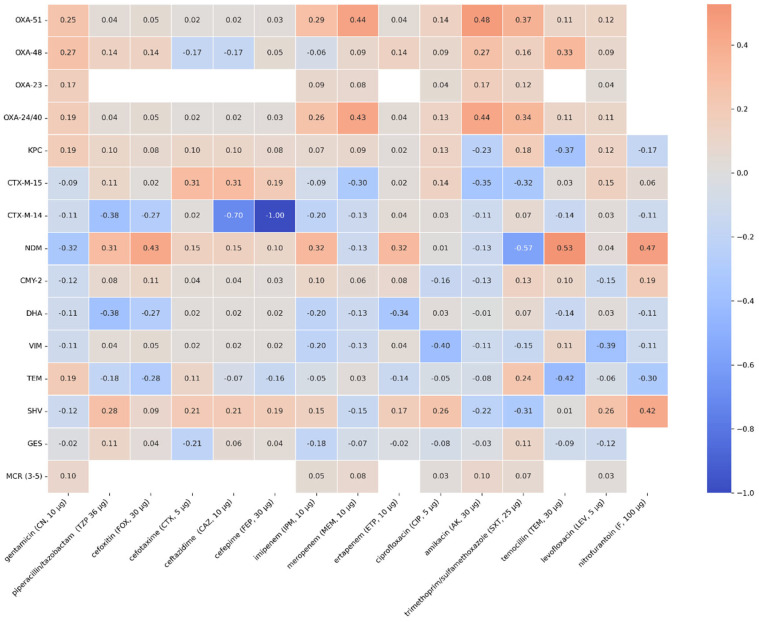
Spearman’s rank correlation coefficients of the genes and phenotypic antimicrobial resistance that were tested.

**Figure 8 antibiotics-14-01275-f008:**
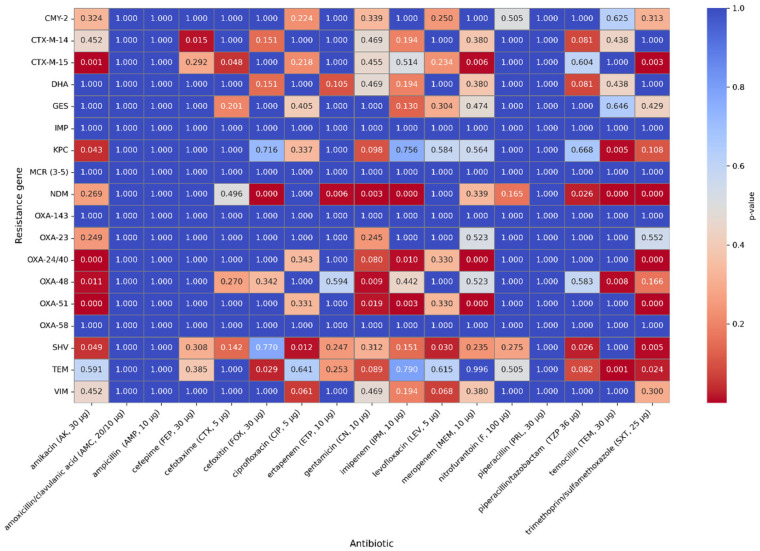
*p*-values of the genes and phenotypic antimicrobial resistance that were tested.

**Figure 9 antibiotics-14-01275-f009:**
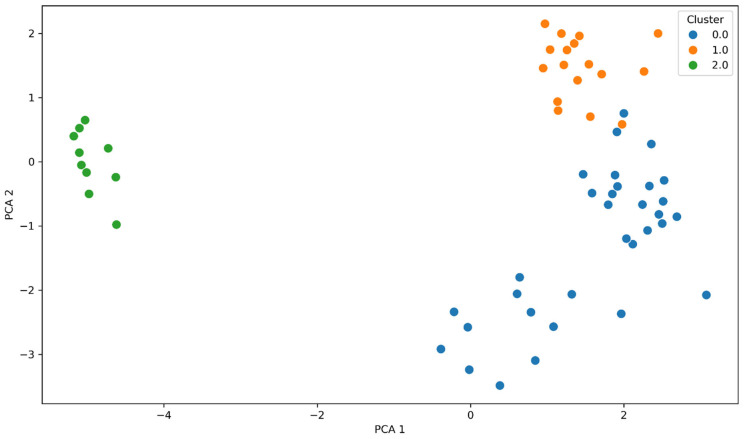
Two-dimensional principal component analysis (PCA) based on standardized genotypic (resistance genes) and phenotypic (S/I/R antibiotic susceptibility categories) profile data. Three clusters are visible, corresponding to different mechanisms and antibiotic susceptibility. Cluster 0 (blue) groups isolates with predominantly ESBL mechanisms, cluster 1 (orange) represents multidrug-resistant strains with multiple resistance genes, and cluster 2 (green) represents strains with dominant genes encoding OXA-type carbapenemases.

**Table 1 antibiotics-14-01275-t001:** Occurrence of genes determining carbapenemase and ESBL resistance among the tested strains.

	*K. pneumoniae* *N* = 63 *n* (%)	*A. baumannii* *N* = 25 *n* (%)	*E. coli* *N* = 13 *n* (%)
Carbapenemase			
NDM	29 (46.03)	0 (0)	5 (38.46)
OXA-48	12 (19.04)	0 (0)	0 (0)
VIM	0 (0)	0 (0)	1 (7.70)
KPC	22 (34.92)	0 (0)	1 (7.70)
OXA-24/40	0 (0)	21 (84.00)	0 (0)
OXA-51	0 (0)	24 (96.00)	0 (0)
OXA-23	0 (0)	3 (12.00)	0 (0)
ESBL			
TEM	26 (41.26)	5 (20.00)	6 (46.15)
SHV	53 (84.12)	0 (0)	0 (0)
CTX-M-15	56 (88.89)	0 (0)	7 (53.85)
		AmpC	
CMY-2	0 (0)	0 (0)	4 (30.77)
DHA	0 (0)	0 (0)	1 (7.70)

*N*—the number of tested strains. *n*—the number of resistant strains.

**Table 2 antibiotics-14-01275-t002:** Number of strains tested, categorized by type of clinical specimen.

Name of the Microorganism	*Klebsiella pneumoniae*	*Acinetobacter baumannii*	*Escherichia coli*
body cavity fluid	1	0	0
blood	18	13	7
blood from puncture site	1	0	0
urine	5	0	0
rectum	25	5	6
pressure ulcer	1	0	0
leg ulcer	0	1	0
bronchoalveolar lavage fluid	9	6	0
wound	3	0	0
**In total**	**63**	**25**	**13**

**Table 3 antibiotics-14-01275-t003:** Antibiotics classes and agents used for antimicrobial susceptibility testing of *Klebsiella pnaumoniae*, *Acinetobacter baumanii* and *Escherichia coli*.

Species	Antibiotic Classes and Agents (Abbreviation, µg)
*Klebsiella pneumoniae*	**Penicillins:** ampicillin (AMP, 10), piperacillin (PRL, 30), piperacillin/tazobactam (TZP, 36), amoxicillin/clavulanic acid (AMC, 20/10)
	**Cephalosporins:** cefoxitin (FOX, 30), cefotaxime (CTX, 5), ceftazidime (CAZ, 10), cefepime (FEP, 30)
	**Carbapenems:** imipenem (IPM, 10), Meropenem (MEM, 10), ertapenem (ETP, 10)
	**Aminoglycosides:** gentamicin (CN, 10), amikacin (AK, 30)
	**Fluoroquinolones:** ciprofloxacin (CIP, 5), levofloxacin (LEV, 5)
	**Other:** trimethoprim/sulfamethoxazole (SXT, 25), Temocillin (TEM, 30)
*Acinetobacter baumannii*	**Carbapenems:** imipenem (IPM, 10), meropenem (MEM, 10)
	**Aminoglycosides:** gentamicin (CN, 10), amikacin (AK, 30)
	**Fluoroquinolones:** ciprofloxacin (CIP, 5), levofloxacin (LEV, 5)
	**Other:** trimethoprim/Sulfamethoxazole (SXT, 25)
*Escherichia coli*	**Penicillins:** ampicillin (AMP, 10), piperacillin (PRL, 30), piperacillin/tazobactam (TZP, 36), Amoxicillin/Clavulanic acid (AMC, 20/10)
	**Cephalosporins:** cefoxitin (FOX, 30), cefotaxime (CTX, 5), ceftazidime (CAZ, 10), cefepime (FEP, 30)
	**Carbapenems:** imipenem (IPM, 10), meropenem (MEM, 10), ertapenem (ETP, 10)
	**Aminoglycosides:** gentamicin (CN, 10), amikacin (AK, 30)
	**Fluoroquinolones:** ciprofloxacin (CIP, 5), levofloxacin (LEV, 5)
	**Other:** trimethoprim/sulfamethoxazole (SXT, 25), temocillin (TEM, 30), nitrofurantoin (F, 100)

**Table 4 antibiotics-14-01275-t004:** The list of resistance genes detectable by the Streck ARM-D Kit.

Test Name Streck ARM-D Kit	Genes Detected in the Test
Streck ARM-D Kit β-Laktamase	IMP-1, NDM, OXA-48, CMY-2, CTX-M-14, CTX-M-15, DHA, VIM, KPC
Streck ARM-D Kit OXA	OXA-143, OXA-48, OXA-24/40, OXA-58, OXA-51, OXA-23
Streck ARM-D Kit TEM/SHV/GES	TEM, SHV, GES
Streck ARM-D Kit ampC	DHA, CMY-2

## Data Availability

The original contributions presented in this study are included in the article.

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
