# Peer review of "Analysis of the Concordance Between the Use of Phenotypic Screening Tests with the β-Lactamase Gene Profile in Selected Gram-Negative Bacteria"

_antibiotics, 2025, doi:10.3390/antibiotics14121275_

Round 1

Reviewer 1 Report

Comments and Suggestions for Authors

In this manuscript, Patrycja Głowacka et al. test multiple bacterial samples from Poland for their antibiotic resistance via chromogenic media, antimicrobial susceptibility testing, and molecular assays. They find that most samples show high resistance to many antibiotics.

Based on the content, I suggest publishing this manuscript in MDPI Antibiotics, but the authors need to make some revisions.

One of the major issues is the Results 3.1. You conclude that the Brilliance CRE and ESBL agars are not effective because all the tested bacteria grow regardless of their phenotype. I'm wondering if there might be potential overgrowth or unintended co-culturing to your samples, as they were collected in military hospitals. In addition, have you tried chromogenic agars from other manufacturers? Would they perform the same as Brilliance's products? It would be great to include experiments using at least one alternative medium.

Another issue is that all "A. baumannii" are misspelled as "A. baumanii".

In Figure 7. When I look into this figure, it seems to me that the cluster 0 should be only composed of the points whose PCA2 values are less than -1.5. I'm wondering if these points contain only the E. coli? If so, the K-means clustering results appear unnecessary to me.

Author Response

Comment 1: One of the major issues is the Results 3.1. You conclude that the Brilliance CRE and ESBL agars are not effective because all the tested bacteria grow regardless of their phenotype. I'm wondering if there might be potential overgrowth or unintended co-culturing to your samples, as they were collected in military hospitals. In addition, have you tried chromogenic agars from other manufacturers? Would they perform the same as Brilliance's products? It would be great to include experiments using at least one alternative medium.

Response 1: All tested isolates were initially identified to the species level in the hospital laboratory using routine diagnostic methods. They were then transported to our Center, where they were replated on Columbia agar to assess culture purity, and then banked at -70°C. Importantly, all studies were conducted in compliance with GLP, which minimizes the risk of contamination. However, as with biological material, this cannot be completely ruled out. We agree that a comparative assessment of alternative media would be valuable; however, in our initial studies, we focused on analyzing commonly available media used in laboratories. The discussion included results presented by other research teams and added a point regarding expanding future studies to include comparative analysis of alternative media.

Comment 2: Another issue is that all "A. baumannii" are misspelled as "A. baumanii".

Response 2: The misspelling has been corrected.

Comment 3: In Figure 7. When I look into this figure, it seems to me that the cluster 0 should be only composed of the points whose PCA2 values are less than -1.5. I'm wondering if these points contain only the E. coli? If so, the K-means clustering results appear unnecessary to me.

Response 3: Cluster 0 is not exclusively composed of E. coli. According to the species proportion, cluster 0 consists of Klebsiella pneumoniae (86%) and Escherichia coli (14%). Cluster 1 consists of K. pneumoniae (76%), E. coli (21%), and A. baumannii (3%). Cluster 2 is composed of A. baumannii. This separation results from differences in resistance gene profiles and antimicrobial resistance patterns, which were variables in the PCA and k-Mears analyses. The distinct cluster formed by A. baumannii results from its characteristic distinct genetic profile formed by OXA carbapenemase genes. Details of cluster compositions are included in the text.

The changes have been marked in red, and the sections we decided to remove from the text have been crossed out.

Reviewer 2 Report

Comments and Suggestions for Authors

The present study was conducted for the purpose of investigating the correlation between the findings of antimicrobial susceptibility testing on ESBL and CRE chromogenic media and the presence of specific resistance genes.

Major revisions:

The title: It must be noted that the title does not meet the requisite standards of clarity. It is imperative that the title is rewritten to ensure that it fulfils the criteria for being both informative and clear.

Line 25: Replace “genetic” with “genotypic” and remove “molecular”

Line 26: in aim of the current study….. antimicrobial susceptibility assessed, with the presence  of specific resistance genes.  It is imperative to define the subjects of the study in question, namely, to specify the chromogenic media and specific resistance genes that will be used in the study's susceptibility testing.

Line 27: replace “following” with “present”…..replace “collected” with “obtained”

Line 29: replace “samples” with “isolates or strains”

Line 30: ….. molecular assays… for what? Which genes..beta-lactam and carbapenem resistance genes..Clarify?

Line 31-32: Remove this sentence “Statistical analysis applied Spearman’s rank correlation coefficients and p-values”.

Line 32: The result remains unclear in abstract, as the objective of the study was to investigate the association between the phenotypic and genotypic for ESBL and CRE resistance. The authors must be written about how much ESBL resistance using chromogenic media and specific ESBL resistance genes was detected. This section requires rewriting.

Line 44: “Profiling of the tested isolates indicated the three clusters with diverse genetic and phenotypic profiles” unclear sentence, it is imperative to rewrite it again.

Line 52: The necessity of the subtitles in the introduction is unclear; however, it is recommended that the style of the journal be followed. It is imperative that the author rewrites the introduction, including a detailed account of the previous methods used to detect resistance, both phenotypically and genotypically.

Line 84: …. (e.g., Phoenix BD [Becton 84 Dickinson], Vitek 2 [bioMérieux]) the country must be included?

Line 111-124: in this paragraph “Due to the serious threat posed by carbapenem-resistant strains, there is particular interest in searching for additional genes encoding this resistance. In the case of carbapenemases, molecular diagnostics largely rely on detecting genes from representatives belonging to three classes: A, B, and D (according to molecular differentiation based on Ambler classification). The first class, class A, also known as serine carbapenemases, includes the KPC gene (Klebsiella pneumoniae carbapenemase). This is one of the most widespread carbapenemases, mainly found in Enterobacterales, and strongly hydrolyzes all β-lactams, including carbapenems. Class B genes belong to MBLs (metallo-β-lactamases dependent on zinc ions), which include NDM (New Delhi metallo-β-lactamase), widely distributed and conferring resistance to all β-lactams except monobactams (e.g., aztreonam), VIM (Verona integron-encoded metallo-β-lactamase) found in Pseudomonas aeruginosa, Enterobacterales, and hydrolyzes  carbapenems, IMP (Imipenem metallo-β-lactamase) present in Pseudomonas aeruginosa and Acinetobacter baumannii, acting similarly to VIM”… the appropriate references must be included. I noticed that the authors cited one reference (No.9), which not appropriate for this paragraph.

Line 140-141: .. Genes responsible for this type of resistance are most often located on mobile genetic elements such as plasmids or transposons. Please cite the appropriate references here.

It is imperative that the limitations of the phenotypic approaches be incorporated into the conclusion of the discussion section.

Line 171: the subtitle “Growth on chromogenic media”(not clear), Given that selective chromogenic media was utilised for the detection of ESBL or CRE resistance, it is imperative that the title is revised to ensure its continued informative value.

Line 173: the order Enterobacterales is italic, while sp. is not italic (Klebsiella sp.). Please check and correct in whole manuscript.

It is my opinion that the presentation of the main findings in the current study was not clear. For this reason, I suggested that the result section should be rewritten. As the materials and methods section was located at the end of the manuscript, it is imperative that the first paragraph of each result title provides a concise overview of the methods employed.

The present study's primary objective is to clearly demonstrate the association between phenotypic and genotypic resistance. This association must be emphasised as a fundamental element of the study's findings.

Line 291: Profiling of the tested isolates and figure 7, it was not clear for? Please clarify?

Line 305/306: in this paragraph …In the analyzed materials, a total of 101 isolates of Gram-negative bacteria were tested, of which the isolated genes 306 representing the ESBL resistance mechanism. The clarity of this paragraph needs to be improved.

Line 312: Enterobacteriaceae is italic. Please remove “strains”.

It was proposed that the discussion section should be rewritten in a clearer manner.

Line 417: samples OR isolates. Clarify?

Line 445: in the the title “Antimicrobial susceptibility testing - disc diffusion methodology” The methods must be rewritten for further clarity, as their presentation is currently lacking in clarity and conciseness.

Line 446: Correct, Kirby’ego “Kirby”..Full name of EUCAST, It is therefore recommended that the authors prepare a list of abbreviations to be included in the current manuscript.

Line 447: 0,5  replace with 0.5.

Line 449: (BIOMAXIMA), (OXOID)…atc, the country should be included in all companies.

Comment: The appropriate references must be included in the materials and methods section.

Line 479: the information of quality control for both methods whether phenotypic or genotypic should be included in the subtitles 5.1 and 5.2.

Comment: It is imperative that the nomenclature of genera, species, orders and families be inscribed in italics within the reference list.

Comments on the Quality of English Language

The English language used in the text could be improved by means of expressing the research in a clearer manner.

Author Response

Comment 1: The title: It must be noted that the title does not meet the requisite standards of clarity. It is imperative that the title is rewritten to ensure that it fulfils the criteria for being both informative and clear.

Response 1: The title has been revised — it has been shortened and made more general : „Analysis of the concordance between the use of phenotypic screening tests with the β-lactamase gene profile in selected Gram-negative bacteria”

Comment 2: Line 25: Replace “genetic” with “genotypic” and remove “molecular”

Response 2: The word “genetic” has been replaced with “genotypic”. The word “molecular” has been removed.

Comment 3: Line 26: in aim of the current study….. antimicrobial susceptibility assessed, with the presence  of specific resistance genes.  It is imperative to define the subjects of the study in question, namely, to specify the chromogenic media and specific resistance genes that will be used in the study's susceptibility testing.

Response 3: The aim of the study has been redefined, and the specific chromogenic media as well as resistance genes used in tests have been specified. The corrected sentence is:

The study aims to determine drug sensitivity and to analyze the correlation between the results obtained from cultures on commercial chromogenic media BrillianceTM CRE (OXOID) and Brilliance TM ESBL (OXOID) and the occurrence of specific resistance genes carbapenemase (IMP, NDM, VIM, KPC, OXA), ESBL β-lactamase (TEM, SHV, CTX-M), and AmpC (CMY, DHA), which will be used in drug sensitivity tests.

Comment 4: Line 27: replace “following” with “present”…..replace “collected” with “obtained”

Response 4: The word “following” has been replaced with “present”. The word “collected” has been replaced with “obtained”.

Comment 5: Line 29: replace “samples” with “isolates or strains”

Response 5: The word “samples” has been replaced with “strains”.

Comment 6: Line 30: ….. molecular assays… for what? Which genes..beta-lactam and carbapenem resistance genes.. Clarify?

Response 6 : The sentence was clarified by listing the names of the genes responsible for resistance mechanisms that were used in the study. The corrected sentence is:

Additionally, molecular assays detecting three main classes according to the mechanism of action and enzyme type carbapenemase (IMP, NDM, VIM, KPC, OXA) and ESBL β-lactamase (TEM, SHV, CTX-M), and AmpC (CMY, DHA) were performed using the real-time PCR method.

Comment 7: Line 31-32: Remove this sentence “Statistical analysis applied Spearman’s rank correlation coefficients and p-values”.

Response 7: The sentence has been removed.

Comment 8 : Line 32: The result remains unclear in abstract, as the objective of the study was to investigate the association between the phenotypic and genotypic for ESBL and CRE resistance. The authors must be written about how much ESBL resistance using chromogenic media and specific ESBL resistance genes was detected. This section requires rewriting.

Response 8: The authors specified detection methods as follows:

Line 43-46: All tested strains grew on chromogenic BrillianceTM CRE medium. In the case of Brilliance TM ESBL medium, the genes determining the resistance mechanism tested were detected in 41.7% for A. baumannii, 53.8% for E. coli, and 100% for K. pneumoniae. Chromogenic media perfectly differentiate strains to species.

Comment 9: Line 44: “Profiling of the tested isolates indicated the three clusters with diverse genetic and phenotypic profiles” unclear sentence, it is imperative to rewrite it again.

Response 9: The sentence has been clarified by adding descriptions of the particular clades. The corrected sentence is:

K-means cluster analysis performed on integrated genotype-phenotype data allowed for the identification of three distinct clusters characterized by distinct resistance gene profiles. Cluster 0 is a heterogeneous cluster grouping strains with ESBL genes, primarily CTX-M (62%). Cluster 1 groups strains with the highest resistance and the highest MDR gene resistance (NDM- 58%, KPC- 30%, CTX-M-15 82%, TEM-55%, GES 55%). Cluster 2 is the most extreme cluster with isolates with predominantly OXA-51 (88%) and OXA-24/40 (77%) genes.

Comment 10: Line 52: The necessity of the subtitles in the introduction is unclear; however, it is recommended that the style of the journal be followed. It is imperative that the author rewrites the introduction, including a detailed account of the previous methods used to detect resistance, both phenotypically and genotypically.

Response 10: The subtitles have been removed to improve clarity of the introduction.

Comment 11: Line 84: …. (e.g., Phoenix BD [Becton 84 Dickinson], Vitek 2 [bioMérieux]) the country must be included?

Response 11: The countries of the entrepreneurs have been included as follows: Phoenix BD (Becton Dickinson, USA), and Vitek 2 (bioMérieux, France).

Comment 12: Line 111-124: in this paragraph “Due to the serious threat posed by carbapenem-resistant strains, there is particular interest in searching for additional genes encoding this resistance. In the case of carbapenemases, molecular diagnostics largely rely on detecting genes from representatives belonging to three classes: A, B, and D (according to molecular differentiation based on Ambler classification). The first class, class A, also known as serine carbapenemases, includes the KPC gene (Klebsiella pneumoniae carbapenemase). This is one of the most widespread carbapenemases, mainly found in Enterobacterales, and strongly hydrolyzes all β-lactams, including carbapenems. Class B genes belong to MBLs (metallo-β-lactamases dependent on zinc ions), which include NDM (New Delhi metallo-β-lactamase), widely distributed and conferring resistance to all β-lactams except monobactams (e.g., aztreonam), VIM (Verona integron-encoded metallo-β-lactamase) found in Pseudomonas aeruginosa, Enterobacterales, and hydrolyzes  carbapenems, IMP (Imipenem metallo-β-lactamase) present in Pseudomonas aeruginosa and Acinetobacter baumannii, acting similarly to VIM”… the appropriate references must be included. I noticed that the authors cited one reference (No.9), which not appropriate for this paragraph.

Response 12: The appropriate reference for this paragraph has been implemented:

Skarżyńska, M.; Zając, M.; Wasyl, D. Antibiotics and bacteria: mechanisms of action and resistance strategies. Adv. Microbiol. 2020, 59 (1), 49–62. https://doi.org/10.21307/PM-2020.59.1.005.

Dzierżanowska-Fangrat, K.; Gniadkowski, M.; Hońdo, Ł.; Hryniewicz, W.; Literacka, E.; Mączyńska, A.; Ozorowski, T.; Papierowska-Kozdój, W.; Pawlik, K.; Wanke-Rytt, M.; Żabicka, D.; Żukowska, A. Carbapenemase - producing Enterobacterales (CPE): epidemiology, diagnosis, treatment, and infection prevention. In: Hryniewicz, W.; Kuch, A.; Wanke-Rytt, M.; Żukowska, A., Eds. Narodowy Instytut Leków, Warsaw, 2022.

Comment 13: Line 140-141: .. Genes responsible for this type of resistance are most often located on mobile genetic elements such as plasmids or transposons. Please cite the appropriate references here.

Response 13: The citation was supplemented by:

Skarżyńska, M.; Zając, M.; Wasyl, D. Antibiotics and bacteria: mechanisms of action and resistance strategies. Adv. Microbiol. 2020, 59 (1), 49–62. https://doi.org/10.21307/PM-2020.59.1.005.

Dzierżanowska-Fangrat, K.; Gniadkowski, M.; Hońdo, Ł.; Hryniewicz, W.; Literacka, E.; Mączyńska, A.; Ozorowski, T.; Papierowska-Kozdój, W.; Pawlik, K.; Wanke-Rytt, M.; Żabicka, D.; Żukowska, A. Carbapenemase - producing Enterobacterales (CPE): epidemiology, diagnosis, treatment, and infection prevention. In: Hryniewicz, W.; Kuch, A.; Wanke-Rytt, M.; Żukowska, A., Eds. National Medicines Institute Narodowy Instytut Leków, Warsaw, 2022.

Garsevanyan, S.; Barlow, M. The Klebsiella pneumoniae carbapenemase (KPC) β-lactamase has evolved in response to ceftazidime avibactam. Antibiotics 2023, 13 (1), 40. https://doi.org/10.3390/antibiotics13010040.

Comment 14: It is imperative that the limitations of the phenotypic approaches be incorporated into the conclusion of the discussion section.

Response 14: The limitations of the phenotypic approaches have been added to the summary of the discussion. The incorporated sentence is: 

Line 460-466: However these methods also have some drawbacks. It is widely known that phenotypic methods have significant limitations. For example, depending on their environment microorganisms can alter their gene expression. In such situations, genetically identical microorganisms may display different phenotypic traits. Point mutations may also occur in bacterial genomes, leading to changes in phenotype. Therefore, phenotypic and genotypic methods complement each other and should be used simultaneously in order to achieve the best possible understanding of the mechanisms of antibiotic resistance [4].

Comment 15: Line 171: the subtitle “Growth on chromogenic media”(not clear), Given that selective chromogenic media was utilised for the detection of ESBL or CRE resistance, it is imperative that the title is revised to ensure its continued informative value.

Response 15: The subtitle “Growth on chromogenic media” has been replaced with “Growth on chromogenic media Brilliance™ CRE and Brilliance™ ESBL”.

Comment 16: Line 173: the order Enterobacterales is italic, while sp. is not italic (Klebsiella sp.). Please check and correct in whole manuscript.

Response 16: All species names have been corrected and put in italics. The order name has been left without italics.

Comment 17: It is my opinion that the presentation of the main findings in the current study was not clear. For this reason, I suggested that the result section should be rewritten. As the materials and methods section was located at the end of the manuscript, it is imperative that the first paragraph of each result title provides a concise overview of the methods employed.

The present study's primary objective is to clearly demonstrate the association between phenotypic and genotypic resistance. This association must be emphasised as a fundamental element of the study's findings.

Response 17: The titles of the results have been edited to emphasize the methods used in the study

Comment 18: Line 291: Profiling of the tested isolates and figure 7, it was not clear for? Please clarify?

Response 18: The following clarifications regarding Figure 7 have been added: Line 329-334: Two-dimensional principal component analysis (PCA) based on standardized genotypic (resistance genes) and phenotypic (S/I/R antibiotic susceptibility categories) profile data. Three clusters are visible, corresponding to different mechanisms and antibiotic susceptibility. Cluster 0 (blue) groups isolates with predominantly ESBL mechanisms, cluster 1 (orange) represents multidrug-resistant strains with multiple resistance genes, and cluster 2 (green) represents strains with dominant genes encoding OXA-type carbapenemases.

Additionally, appropriate explanations have been incorporated to the text of section 3.5 Profiling of the tested isolates. The incorporated text is: 

Line 305-327: K-means cluster analysis performed on integrated genotype-phenotype data revealed the presence of three distinct resistance profiles, characterized by different combinations of carbapenemase and beta-lactamase genes and varying levels of antibiotic resistance. Cluster 0 is characterized by a moderate presence of ESBL genes, primarily CTX-M-15 (62%), and a low prevalence of class A (KPC 19%) and class D (OXA-48 6%) carbapenemases. Phenotypically, these strains exhibited low levels of aminoglycoside resistance (AK: 12%) and lower levels of carbapenem resistance (MEM: 0%, IMP: 6%) compared to the other clusters. This cluster exhibited high levels of cephalosporin resistance (FEP: 94%, CAZ: 94%). Cluster 1 is characterized by the highest prevalence of metallo-beta-lactamases (NDM 58%), and class A and D carbapenemases, i.e., KPC (30%) and OXA-48 (17%). Furthermore, there is a high prevalence of the ESBL gene CTX-M-15 (82%). The presence of AmpC CMY-2, TEM (55%), and TEM (55%) has also been observed. Strains grouped in this cluster are characterized by the highest levels of resistance to carbapenems (MEM: 57%, IMP: 97%), and broad resistance to aminoglycosides (AK: 48%), fluoroquinolones, and cephalosporins (FEP: 76%, CAZ: 83%). These strains have an MDR profile. These strains constitute the most clinically problematic group with limited therapeutic options. Cluster 2 is the cluster with the dominant OXA-51 (88%) and OXA-24/40 (77%) genes. A low presence of class A carbapenemase genes (KPC 4%) was observed, and ESBL, AmpC, and metallo-β-lactamases were absent in this cluster. These strains phenotypically exhibited high-level resistance to carbapenems (MEM: 88%, IMP: 92%) and aminoglycosides (AK: 88%), and low-level resistance to cephalosporins (FEP: 11%, CZA: 12%). Resistance to quinolones was high in all clusters: cluster 0 81%, cluster 1 88%, cluster 2 88%.

Comment 19: Line 305/306: in this paragraph …In the analyzed materials, a total of 101 isolates of Gram-negative bacteria were tested, of which the isolated genes 306 representing the ESBL resistance mechanism. The clarity of this paragraph needs to be improved.

Response 19: The paragraph has been made more specific: In the analyzed materials, a total of 101 isolates of Gram-negative bacteria were tested, of which the isolated genes representing the ESBL resistance mechanism were CTX-M-15 (56), SHV (53), and TEM (26) in K. pneumoniae, TEM (6) and CTX-M-15 (7) in E. coli and TEM (5) in A. baumanii.

Comment 20: Line 312: Enterobacteriaceae is italic. Please remove “strains”.

It was proposed that the discussion section should be rewritten in a clearer manner.

Response 20: The discussion has been modified.

Comment 21: Line 417: samples OR isolates. Clarify?

Response 21: Samples collected from patients were identified in the hospital laboratory and then a single pure isolate of the microorganism was transferred to the Center.

Comment 22: Line 445: in the the title “Antimicrobial susceptibility testing - disc diffusion methodology” The methods must be rewritten for further clarity, as their presentation is currently lacking in clarity and conciseness.

Response 22: The description of the antibiotics used to perform antibacterial antibacterial tests using the disk diffusion method has been modified. Antibiotics have been divided into groups and assigned to microorganisms in a table, which will improve the clarity of the presented data.

Comment 23: Line 446: Correct, Kirby’ego “Kirby”..Full name of EUCAST, It is therefore recommended that the authors prepare a list of abbreviations to be included in the current manuscript.

Response 23:  The corrected “Kirby-Bauer” form and the full name of EUCAST (the European Committee on Antimicrobial Susceptibility Testing) have been replaced.

Comment 24: Line 447: 0,5  replace with 0.5.

Response 24:  The “0,5” has been replaced with “0.5”.

Comment 25: Line 449: (BIOMAXIMA), (OXOID)…atc, the country should be included in all companies.

Response 25: The countries of the entrepreneurs have been included in the materials and methods section as follows: (BIOMAXIMA, Poland), (OXOID, Germany) – in terms of chromogenic media, (OXOID, UK) – in terms of discs containing antibiotics, and (SaMag-12, Sacace, Italy).

Comment 26: The appropriate references must be included in the materials and methods section.

Response 26: In the materials and methods section the appropriate reference has been included: 

29. European Committee on Antimicrobial Susceptibility Testing. (2025). Breakpoint tables for interpretation of MICs and zone diameters, version 15.0. http://www.eucast.org/clinical_breakpoints.

The description of materials and methods was prepared based on the instructions provided by the manufacturers of the research materials used.

Comment 27: Line 479: the information of quality control for both methods whether phenotypic or genotypic should be included in the subtitles 5.1 and 5.2.

Response 27: The appropriate quality control information for chromogenic media and molecular methods RT-PCR assay has been placed in the subtitles 5.1 and 5.2.

Comment 28: It is imperative that the nomenclature of genera, species, orders and families be inscribed in italics within the reference list.

Response 28: Nomenclature has been corrected.

The changes have been marked in red, and the sections we decided to remove from the text have been crossed out.

Round 2

Reviewer 1 Report

Comments and Suggestions for Authors

The revised manuscript and the authors’ responses satisfy my concerns. I now recommend publishing the manuscript in present form.

Author Response

Comment 1: Line 47-48: correct…. the occurrence of the antibiotic resistance genes was observed for OXA-51 and OXA-24/40 genes, which resistance to meropenem..

Response 1: The sentence has been corrected according to the guidelines.

Comment 2: Line 51_54: This sentence is not required in this context and, consequently, should be removed in order to reduce the length of the abstract., …. Cluster 0 is a heterogeneous cluster grouping strains with ESBL genes, primarily CTX-M (62%). Cluster 1 groups strains with the highest resistance and the highest MDR gene resistance (NDM- 58%, KPC- 30%, CTX-M-15 82%, TEM-55%, GES 55%). Cluster 2 is the most extreme cluster with isolates with redominantly OXA-51 (88%) and OXA-24/40 (77%) genes.

Response 2: The sentence has been removed.

Comment 3: Line 99: the full name of ESBLs should be included for first time.

Response 3: The full name of ESBLs has been included.

Comment 4:Line 146: remove! (Extended-Spectrum β-Lactamases), since the full name was written above.

Response 4: The full name of ESBLs has been removed.

Comment 5: Response 10: The subtitles have been removed to improve clarity of the introduction. The subtitles were note removed based on the journal style.

Response 5: The subtitle from the introduction has been removed. The following number of subtitles has been corrected according to the order.

Comment 6: Line 182: Enterobacteriaceae should be italicized.

Response 6: The term Enterobacteriaceae has been corrected to italicized.

Comment 7: Line 232: K. pneumoniae is italicized.

Response 7: The term K. pneumoniae has been corrected to italicized.

Reviewer 2 Report

Comments and Suggestions for Authors

Line 47-48: correct…. the occurrence of the antibiotic resistance genes was observed for OXA-51 and OXA-24/40 genes, which resistance to meropenem..

Line 51_54: This sentence is not required in this context and, consequently, should be removed in order to reduce the length of the abstract., …. Cluster 0 is a heterogeneous cluster grouping strains with ESBL genes, primarily CTX-M (62%). Cluster 1 groups strains with the highest resistance and the highest MDR gene resistance (NDM- 58%, KPC- 30%, CTX-M-15 82%, TEM-55%, GES 55%). Cluster 2 is the most extreme cluster with isolates with redominantly OXA-51 (88%) and OXA-24/40 (77%) genes.

Line 99: the full name of ESBLs should be included for first time.

Line 146: remove! (Extended-Spectrum β-Lactamases), since the full name was written above.

Response 10: The subtitles have been removed to improve clarity of the introduction. The subtitles were note removed based on the journal style.

Line 182: Enterobacteriaceae should be italicized.

Line 232: K. pneumoniae is italicized.

Author Response

(The authors gave the same response as above.)
